# In-context learning in presence of spurious correlations

**Hrayr Harutyunyan** [1]   **Rafayel Darbinyan** [2 3]   **Samvel Karapetyan** [2 4]   **Hrant Khachatrian** [2 4]

## Abstract

Large language models exhibit a remarkable capacity for in-context learning, where they learn to solve tasks given a few examples. Recent work has shown that transformers can be trained to perform simple regression tasks in-context. This work explores the possibility of training an in-context learner for classification tasks involving spurious features. We propose a novel technique to train such a learner for a given classification task. Remarkably, this in-context learner matches and sometimes outperforms strong methods like ERM and GroupDRO. However, unlike these algorithms, it does not generalize well to other tasks. We show that it is possible to obtain an in-context learner that generalizes to unseen tasks by constructing a diverse dataset of synthetic in-context learning instances.

## 1. Introduction

Large language models, such as GPT-3, have the ability of in-context learning (ICL), wherein they learn to solve a task given a few examples in the context (Brown et al., 2020). The most significant aspect of in-context learning is that the learning happens during the forward pass on the context and query, without updating network parameters. In order to study in-context learning in isolation, a number of studies considered training transformers (Vaswani et al., 2017) from scratch to solve simple learning tasks in-context. In particular, Garg et al. (2022) show empirically that transformers can be trained to perform in-context learning of simple regression functions, such as dense or sparse linear functions, two-layer ReLU neural networks, and small decision trees.

Training on ICL instances can be seen as an instance of meta-learning (Schmidhuber, 1987; Naik & Mammone, 1992; Thrun & Pratt, 1998), where the goal is to learn a learning

---

[1]Google Research NYC [2]YerevaNN [3]ServiceTitan [4]Yerevan State University. Correspondence to: Hrayr Harutyunyan <hrayrh@google.com>.

*Proceedings of the 1$^{st}$ Workshop on In-Context Learning at the 41$^{st}$ International Conference on Machine Learning*, Vienna, Austria. 2024. Copyright 2024 by the author(s).

algorithm. What exact algorithm is learned when training transformers on ICL instances is still an open problem. Akyürek et al. (2022) and Von Oswald et al. (2023) show that transformers can implement a single gradient descent step of ordinary least squares and update the closed-form solution of ridge regression when a new example is added. Additionally, they provide evidence that transformers trained on ICL instances of linear regression learn algorithms that closely match predictions of the known algorithms, such as gradient descent on ordinary least squares objective and ridge regression. However, there is evidence that the learned algorithm may vary with model scale, depth, and pretraining task diversity (Akyürek et al., 2022; Raventós et al., 2024). In particular, Raventós et al. (2024) demonstrate that in the setting of in-context learning of linear regression tasks with insufficient pretraining task diversity, the learned algorithm behaves like a Bayesian estimator with the pretraining task distribution as the prior, and hence fails to generalize well to unseen tasks. Yadlowsky et al. (2023) show that when trained on ICL instances where the regression function belongs to a union of distinct function classes, the learned algorithm fails to generalize beyond the pretraining function classes. (Ahuja & Lopez-Paz, 2023) show that in-context learning ability diminishes under strong distribution shifts.

In this work, we explore the limits of in-context learning further by testing it on more challenging settings. We deviate from the existing literature and consider visual classification tasks instead of regression tasks with simple function classes. In particular, we consider classification tasks where some features are *spuriously correlated* with the label. Such features are predictive of the label but are not causally related to it, due to which their correlation might not hold at test time. A prominent example is the cow vs camel classification task, where the background often correlates with the label, as cows are typically photographed in pastures, while camels are typically photographed in deserts (Beery et al., 2018). It's well-known that neural networks trained with gradient-based methods to minimize empirical risk can exploit spurious features, causing performance degradation under distribution shifts affecting these correlations (Torralba & Efros, 2011; Ribeiro et al., 2016; Gururangan et al., 2018; Zech et al., 2018; McCoy et al., 2019; Geirhos et al., 2019; 2020; Xiao et al., 2021).

We begin by generating in-context learning instances for a

single task and discuss a few ways of training an in-context learner that is robust spurious features. We find that the simple approach of training an in-context learner explored in the literature leads to models that do classification ignoring the context. Furthermore, these models lack robustness to the spurious feature. We show that the first issue can be mitigated greatly by randomly permuting input embedding dimensions for each training sequence. To address the second issue, we propose a novel way of forming ICL instances and a suitable transformer architecture. The proposed approach outperforms strong baselines such as 1-NN, empirical risk minimization (ERM), and GroupDRO (Sagawa* et al., 2020). However, it does not generalize to new tasks, as all training instances are derived from a single task.

We next explore training an in-context learner that generalizes to unseen tasks with spurious features. We create a dataset of in-context learning instances for various binary classifications tasks with varying spurious features. We demonstrate the efficacy of the proposed approach on this dataset too and find that it can be improved further by passing spurious feature annotations as input and injecting occasional queries requesting the label of a proceeding context example to promote learning induction heads. The resulting model generalizes perfectly to unseen tasks, as long as the data generating process is similar. However, generalization to unseen tasks with possibly different data generating process depends on the severity of the challenge posed by spurious features, indicating that the learned algorithm is more brittle to severe distribution shifts than conventional algorithms. The source code for reproducing our experiments is available at `anonymized`.

## 2. In-context learning based on a single task

For simplicity, we focus on in-context learning of binary classification tasks in presence of a single binary spurious feature. Furthermore, we focus on the case where both classes are equally represented in the training set, although everything in subsequent parts of this work applies to class-imbalanced settings too. Training a transformer to perform linear regression in-context requires millions of ICL training instances, even for small dimensional cases. For example, Garg et al. (2022) use 32 million training instances for 20-dimensional inputs. To generate so many ICL instances, we can take an existing classification task with spurious features and construct many ICL instances from it.

Let $\mathcal{D}_{\text{train}}$ be a set of training examples for the task, where each example is a triplet $(x, s, y)$ of input $x \in \mathbb{R}^d$, spurious feature value $s \in \{0, 1\}$, and label $y \in \{0, 1\}$. Similarly, let $\mathcal{D}_{\text{test}}$ be a set of test examples. Importantly, we do not make any assumptions on the data generating process, except that $x$ has some information about $s$ and $s$ is predictive of $y$ on the training set, but their correlation does not hold on the test

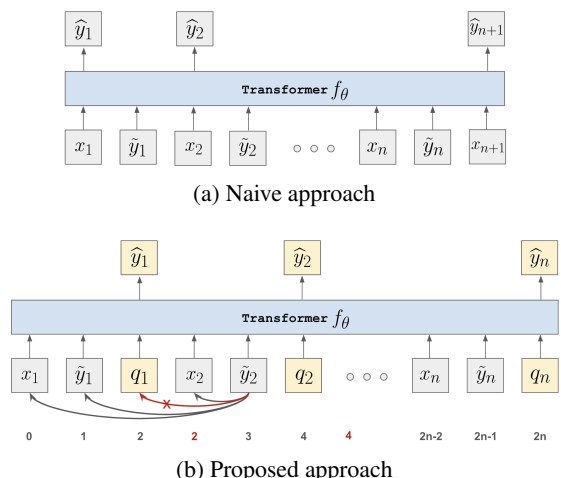

(a) Naive approach

(b) Proposed approach

Figure 1: In-context learning transformer architectures of the naive and proposed approaches. The proposed approach allows arbitrary query tokens after each learning example. Token positions and attention mask are modified so that these intermediate queries have no effect on other tokens.

set. For an example $(x, s, y)$, we define its *group* $g = 2y+s$. In a binary classification task with a single binary spurious feature, there are four groups. Without loss of generality, we assume that for a majority of training examples we have that $y = z$. Hence we refer to groups 0 and 3 as majority groups, while referring to groups 1 and 2 as minority groups.

### 2.1. A naive approach of constructing ICL instances

To construct a single ICL instance, we can sample a subset of $n+1$ examples $\{(x_i, s_i, y_i)\}_{i=1}^{n+1}$ from $\mathcal{D}_{\text{train}}$ and form a sequence $S$ of form: $S = (x_1, \tilde{y}_1, x_2, \tilde{y}_2, \ldots, x_n, \tilde{y}_n, x_{n+1})$, where $\tilde{y}_i \in \mathbb{R}^d$ is a fixed random representation of either $y_i$ or $g_i$ (this distinction will be elaborated later). Then we can train a transformer $f_\theta : \cup_k \mathbb{R}^{k \times d} \to [0, 1]$ to predict $y_i$ given $S_i \triangleq (x_1, \tilde{y}_1, \ldots, x_{i-1}, \tilde{y}_{i-1}, x_i)$ (see Figure 1a) optimizing the following loss function:

$$\frac{1}{n+1} \sum_{i=1}^{n+1} \text{CE}(y_i, f_\theta(S_i)), \qquad (1)$$

where $\text{CE}(y, \hat{y}) = -y \log \hat{y} - (1 - y) \log(1 - \hat{y})$ is the binary cross-entropy loss.

We explore two options of setting $\tilde{y}_i$. In the first option, we set $\tilde{y}_i$ to represent $y_i$ with a constant vector or its negative in $\mathbb{R}^d$. In this case we aim to obtain an in-context learner that is robust to spurious features without receiving spurious feature annotations as input. ERM is one such learner that minimizes average loss on training examples and does not require spurious feature annotations. In the second option, we set $\tilde{y}_i$ to represent $g_i$ as a sum of two constant vectors

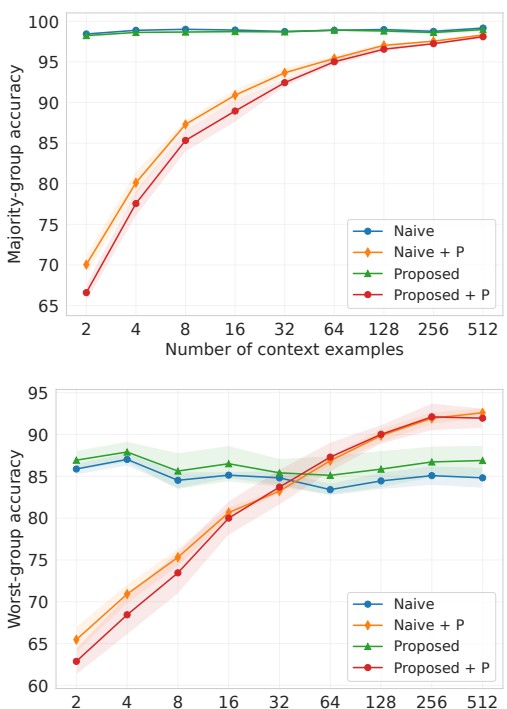

Figure 2: Majority-group and worst-group test accuracies on `Waterbirds` as a function of context size for the naive and proposed approaches with or without permuting input dimensions. Shaded regions show standard deviation across 5 training runs.

in $\mathbb{R}^d$, one representing the class and the other representing the spurious feature. In this case we aim to obtain an in-context learner that does robust classification with respect to a specified spurious feature. GroupDRO is one such learner that minimizes worst-group loss, therefore requiring spurious feature annotations.

Unfortunately, the simple approach of (1) has several issues. First, as the classification task remains the same from one ICL instance to another, the model can learn to ignore context examples and predict $y_i$ based solely on $x_i$. Second, as all $n + 1$ examples of a sequence $S$ are sampled from the training set and the spurious correlation holds for all of them, there is nothing preventing usage of spurious features in making predictions. To confirm these two issues, we consider the `Waterbirds` dataset (Sagawa* et al., 2020), which is landbird vs waterbird image classification task where image background (sea or land) is correlated with the label in the training set (4,795 examples), but not in validation and test sets. A robust classifier should predict `waterbird` or `landbird` without relying on image background. To separate out the representation learning challenge, we represent images with a pretrained and frozen DINOv2 ViT-B/14 distilled (Oquab et al., 2023).

This way each image is embedded in $\mathbb{R}^{768}$. While using powerful pretrained representations increases overall performance under distribution shifts (Radford et al., 2021; Mehta et al., 2022), we note that it does not eliminate the problem of spurious correlations. Representations obtained via large-scale self-supervised pretraining are likely rich enough to capture information about both the label and spurious feature. Furthermore, many works have indicated that the main contribution to the out-of-domain generalization error comes from the classification head (rather than the representation learning module) and called for designing better methods of training the classification head (Galstyan et al., 2022; Menon et al., 2021; Kirichenko et al., 2023; Izmailov et al., 2022; Shi et al., 2023).

We train a causal decoder-only GPT-J transformer (Wang & Komatsuzaki, 2021) with 80M parameters on 2M in-context learning sequence with $n = 512$ and $\tilde{y}_i$ representing labels, constructed from the training set of `Waterbirds`. We use balanced sampling of classes and set the minority group proportion to 10% within each class. We use the ADAM optimizer (Kingma & Ba, 2014) ($\beta_1 = 0.9$ and $\beta_2 = 0.999$) with 32 batch size and no weight decay. The learning rate is selected from $\left\{3 \cdot 10^{-5}, 6 \cdot 10^{-5}, 10^{-4}\right\}$ based on average test performance over 5 runs. Concretely, we evaluate on 8192 sequences where the context part is $n$ training examples, while the query is a sampled from the test set with equal group distribution. Exact metric definitions and missing details are provided in Appendix A. Note that with 512 context length and 10% minority group ratio within each class, the expected value of the number of context examples from each of the 2 minority groups is about 25. For reference, the smallest minority-group has only 56 examples in the `Waterbirds` training set.

Figure 2 plots majority-group and worst-group test accuracies as a function of context size $n$. We see that naive approach results in models that ignore context – worst-group accuracy with 512 context examples is essentially the same as with 2 examples (see the *naive* curve). This validates the first of aforementioned issues. Furthermore, Figure 2 shows that majority-group test accuracy of the naive approach is considerably higher compared to worst-group accuracy, confirming the second issue.

To address the first issue, we propose to rotate image embeddings in each ICL instance independently, making it harder to memorize individual examples. We found that generating random rotation matrices on fly is computationally expensive and slows down training. We tried generating and storing 10K rotation matrices, but this resulted in less than 50M different training examples that were still possible to memorize to some extent. A more effective and efficient alternative is to apply random permutations to image embedding dimensions (for brevity, this technique is denoted

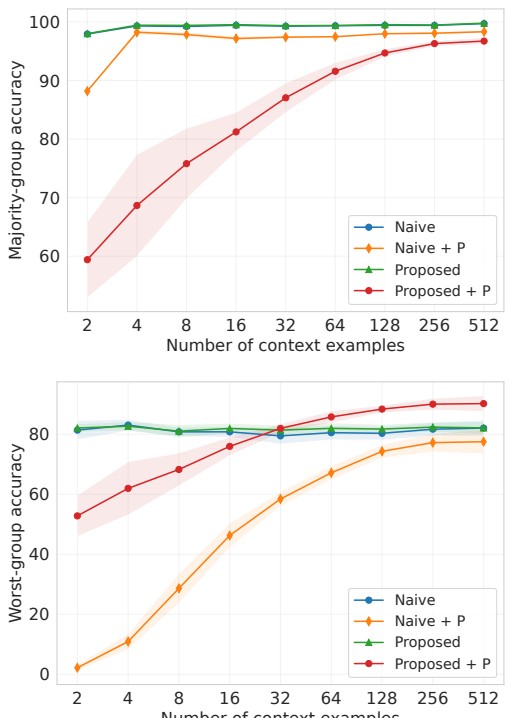

Figure 3: Majority-group and worst-group test accuracies on `Waterbirds-severe` as a function of context size for the naive and proposed approaches with or without permuting input dimensions.

with +$P$ in figures and tables). We found this approach to be very effective in terms of inducing in-context learning (see *naive + P* in Figure 2). We also see that the difference between majority-group and worst-group accuracies decreases, although an approximately 5 p.p. gap remains.

### 2.2. The proposed approach of constructing ICL instances

When training an ICL transformer, ideally, we would like to simulate the situation of making a test prediction based on a context of training examples. Importantly, we would like to simulate the case where test distribution has balanced groups (i.e., the spurious correlation does not hold). Given access to spurious feature annotations for the training set, we can simulate this scenario using only training examples. In particular, we can form ICL instances of form $(x_1, \tilde{y}_1, \ldots, x_n, \tilde{y}_n, x_{n+1})$, where the context examples $(x_1, \ldots, x_n)$ are sampled in a way that the spurious feature is correlated with the label, while the query $x_{n+1}$ is sampled to have a uniform group distribution. However, if we again optimize the loss of (1), for context lengths less than $n$, the network will be allowed to make predictions using the spurious feature. Please refer to Figure 11

of Appendix B for evidence of this. Potential ways of addressing this issue is upweighting the final prediction loss in Eq. (1) or upweighting predictions on minority examples. In our preliminary experiments we found the former approach ineffective. We did not experiment with the latter approach.

Instead, we propose a novel way of forming in-context learning instances and a modified transformer architecture that is suitable for such sequences. In particular, we form sequences of form $S = (x_1, \tilde{y}_1, q_1, x_2, \tilde{y}_2, \ldots, x_n, \tilde{y}_n, q_n)$, where $(x_i, \tilde{y}_i)$ are context examples as before, while $q_i$ are queries, sampled with replacement from $\mathcal{D}_{\text{train}}$ excluding context examples. Importantly, $q_i$ are sampled to have a balanced distribution of groups. Redefining $S_i = (x_1, \tilde{y}_1, q_1, \ldots, x_i, \tilde{y}_i, q_i)$, we would like the final prediction on $S_i$ to be the label of $q_i$. When making a prediction on $q_i$, we want $q_j$ $(j < i)$ to have no effect. Thus, we modify the transformer architecture to disallow attending to query tokens (unless a query token is attending to itself). Furthermore, we modify token positions to discount query tokens. More formally, for the sequence $(x_1, \tilde{y}_1, q_1, x_2, \tilde{y}_2, \ldots, x_n, \tilde{y}_n, q_n)$, positions are a set to $(0, 1, 2, 2, 3, 4, 4, \ldots, 2n-2, 2n-1, 2n)$. Please refer to Figure 1 for an illustration. This is our main approach and will be referred to as "proposed approach" hereafter.

Figure 2 compares the proposed and naive approaches with and without input dimension permutations. Without random permutations, the proposed approach outperforms the naive approach marginally. However, the same is not true with random permutations. We found that image embeddings of DINOv2 have a bias towards representing objects more than backgrounds, alleviating the challenge posed by the spuriously correlated background in `Waterbirds`. For this reason, we create a modified version of Waterbirds by adding a constant vector $s$ or $-s$ to image embeddings based on the spurious feature. We scale $s$ to have its norm equal to the average norm of image embeddings. On this modified waterbirds dataset, which we name `Waterbirds-severe`, we see a large separation between the naive and proposed approaches (see Figure 3). We also see that without permutations, both naive and proposed approaches perform identically, indicating no robustness to the spurious correlation. This is expected, because in the absence of in-context learning, we can think of the naive and proposed approaches, as standard and reweighted empirical risk minimization with a complex classification head, respectively. It has been observed that sample reweighting is not effective in overparameterized settings as all training examples will be perfectly fitted (Byrd & Lipton, 2019; Menon et al., 2021).

### 2.3. Comparison with conventional learning algorithms

Now that we have established the efficacy of the proposed approach with permuted input dimensions, we compare

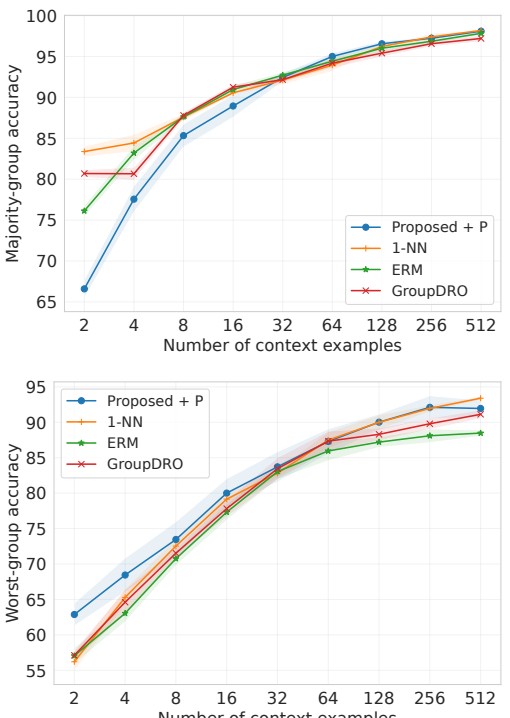

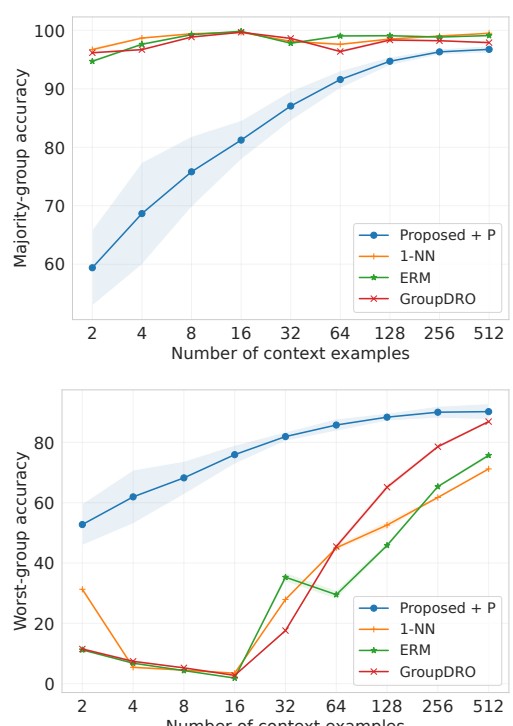

Figure 4: Majority-group and worst-group test accuracies on `Waterbirds` for the proposed approach and baseline methods such as 1-NN, ERM, and GroupDRO.

Figure 5: Majority-group and worst-group test accuracies on `Waterbirds-severe` for the proposed approach and baseline methods such as 1-NN, ERM, and GroupDRO.

it to a few strong algorithms, such as 1-NN, ERM, and GroupDRO. We follow the evaluation recipe used for the in-context learners. Namely, we evaluate each baseline on 8192 sequences by training on the context part of the sequence and making a prediction on the single query. More information about hyperparameters and model selection is presented in Appendix A.

Figures 4 and 5 compare the proposed and baseline approaches on `Waterbirds` and `Waterbirds-severe` respectively. On `Waterbirds`, the proposed method outperforms ERM and GroupDRO on almost all context lengths, but is better than 1-NN only for short context lengths. The good performance of 1-NN is due to the bias in DINOv2 representations. On `Waterbirds-severe`, the proposed method outperforms the baselines at all context lengths. From these results, we conclude that this in-context learner implements none of these algorithms.

It is worth noting that baseline worst-group accuracies at context length $n = 512$ are actually *higher* than what we get when training on the entire dataset. For example, on `Waterbirds`, 1-NN gets only 90.03 worst-group accuracy, while ERM gets $84.23 \pm 0.17$ and GroupDRO gets $92.43 \pm 0.24$. This is due to balanced sampling of classes and setting the minority ratio to 10% withing each class, which

is higher than the minority ratio of $\approx 5\%$ in the original `Waterbirds` dataset. One can think of the resampling we do as a weaker form of down-sampling which has been found to be helpful in presence of spurious correlations (Nagarajan et al., 2021; Menon et al., 2021; Idrissi et al., 2022).

### 2.4. Generality of the learned algorithm

Since we train in-context learners on ICL instances of a single task, a natural question arises whether the learned algorithm can generalize to unseen tasks. Without permuting input dimensions, the network does not learn to do in-context learning. Therefore, we can hope for some generality only when permuting input dimensions. We take a model trained with the proposed method and permutations, and probe generality of its in-context learning by evaluating on various datasets. We start by swapping the two classes in `Waterbirds` at evaluation and observe $\approx 2$ p.p. overall accuracy drop and $\approx 5$ worst-group accuracy drop. Despite the worsened performance, this indicates that the model treats class labels symbolically, which is remarkable given that the semantics of labels were constant during training. When during evaluation we switch the task to predicting background (now the class becomes a spurious feature), the overall test accuracy drops to 54.4%, while the worst-group

accuracy drops to 9.3%. More interestingly, when we evaluate on `Waterbirds-severe`, it gets 100% accuracy on the majority groups and 0% accuracy on minority groups.

However, it is worth noting that the learned algorithm is not completely useless for other tasks and works well when there are no spurious features, even on unseen tasks. For example, evaluating on binary classification tasks derived from the `CUB-200` (Welinder et al., 2010) dataset, from where the bird images of `Waterbirds` were taken, we get 99.7% accuracy at context size 100 (the accuracy is so high because most pairs of classes are easy to distinguish). We also tested on binary classification tasks derived from classes belonging to *Amphibia* and *Mammalia* supercategories of the `iNaturalist` (Van Horn et al., 2018) dataset. At context length 512, the overall accuracy is 98.5%.

These OOD evaluation results indicate that the learned algorithm does something specific to the spurious feature of `Waterbirds`. We hypothesize that it learns to ignore this particular spurious feature. To test this, we evaluate on *group-balanced* `Waterbirds` sequences, with the task set to predicting background, and get 58.5% overall accuracy and 41.3% worst-group accuracy. One potential way of improving generality and possibly also performance, is passing example groups as input, i.e., setting $\tilde{y}_i$ to represent $g_i$. We did not observe performance improvements and increase of generality of the learned algorithm when passing groups as input (see the complete results in Tables 1 and 2 of Appendix B). Thus, we conclude that when all ICL instances are derived from the task, the learned algorithm is inherently tied to the spurious feature of that task.

## 3. In-context learning based on a diverse set of tasks

In Section 2, we showed that it is possible to obtain a good in-context learner for a given task, but it fails to generalize to tasks with different spurious features. A better in-context learner should detect spurious features from context and make predictions without employing them. In this section, we explore the possibility of obtaining such a learner by training on a diverse set of ICL tasks. Since there exist few suitable datasets, we synthesize binary classification tasks with a single binary spurious feature, aiming to capture "structure" present in existing datasets. In short, given a standard binary classification task, say cat vs dog classification, for a sampled minority of cats we overwrite some of their features with those of random dogs. Similarly, we do an analogous operation for a sampled minority of dogs. This way some cats share dog features and vice versa. To create a diverse pool of in-context learning instances, we vary the two classes and the subset of grafted features.

More concretely, we consider the `iNaturalist`

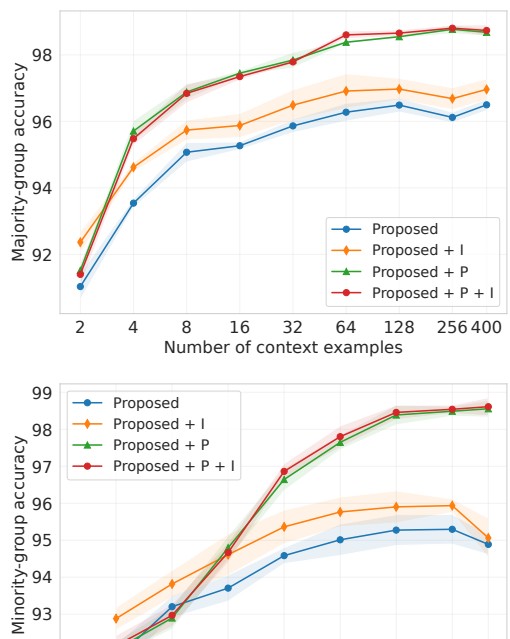

Figure 6: Majority-group and minority-group accuracies on the OOD test set of `iNaturalist` for the proposed approaches with or without permuting input dimensions and promoting induction heads.

dataset (Van Horn et al., 2018), which contains images from 5,089 natural fine-grained categories and filter out categories that have less than 500 images. For testing purposes, from remaining 239 categories we set apart categories that belong to the super-categories *Amphibia* and *Mammalia*, along with 10% of random categories. We denote the set of these 48 categories as $\mathcal{C}_{ood}$, and the set of remaining 191 categories as $\mathcal{C}_{id}$, which we use to create in-context learning instances for training. For each category in $\mathcal{C}_{id}$, we hold out half of the examples as in-distribution validation set. To generate a single in-context learning instance, we sample two distinct classes from $\mathcal{C}_{id}$ randomly and sample $n/2$ images from the training split of each class uniformly at random without replacement. We then do the grafting operation, setting minority group ratio within each class to 10%. We select the grafted features randomly, by first picking subset size $k$ uniformly at random from 0 to 199, and then sampling a random subset of embedding dimensions of size $k$. With this we get $n$ examples that form the context part of the instance. Abandoning the naive approach and focusing on the proposed one, for each class we sample $n/2$ queries from the remaining examples uniformly at random with replacement and do the grafting operation with 50% minority group ratio.

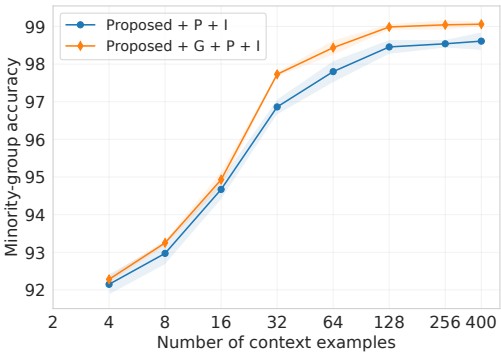

Figure 7: Minority-group accuracy on the OOD test set of `iNaturalist` for the best proposed approach with or without passing group information as input.

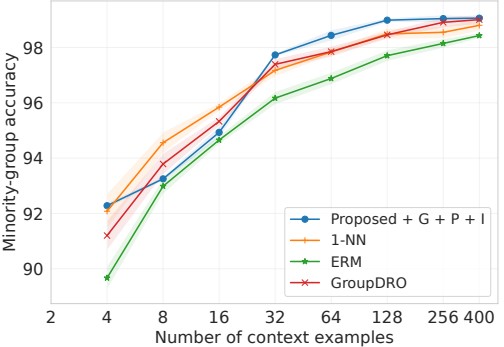

Figure 8: Minority-group accuracy on the OOD test set of `iNaturalist` for the best variant of proposed approach and baseline methods such as 1-NN, ERM, and GroupDRO.

Following the experiments in Section 2, we train the same decoder-only transformer with the proposed architecture on 4M ICL instances with $n = 400$ context examples. We use the same optimizer and sweep the learning rate in the same range, selecting the best value based on the average *minority-group accuracy* (defined exactly in Appendix B) on instances where both categories belong to $\mathcal{C}_{\text{ood}}$ and thus were not observed during training. The results presented in Figure 6 indicate a major difference compared to the results in the single-task regime – namely, the proposed approach learns to do in-context learning to some extent without permuting embedding dimensions. As expected, we see much better performance with permuted embedding dimensions. Notably, comparing majority-group and minority-group accuracies of the proposed approach with permutations, we see almost no sign of reliance on spurious features.

**Promoting emergence of induction heads.** In-context learning ability has been linked to induction heads, which are specific type of circuits found within large language models that implement the operation of looking back over the sequence for finding previous instances of the current token and copying what comes after that (Olsson et al., 2022). Inspired by this, we propose a data preparation technique that promotes learning of induction heads. With probability $p$, we replace each intermediate query independently with a random example from the proceeding part of the context. Note that this type of "hinting" is not possible in the naive approach and is enabled by the introduction of intermediate queries. In all experiments with this technique enabled, we just set $p = 0.25$. We observed that training of typical runs escapes the initial loss plateau faster with this technique (in about 3k iterations compared instead of about 10k iterations). Moreover, we see modest performance gains in `iNaturalist` experiments (see Figure 6, where *+I* stands for this technique).

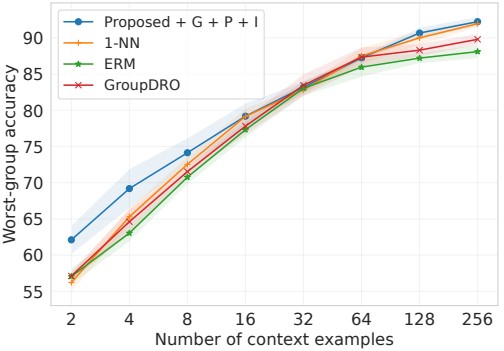

Figure 9: Worst-group test accuracy on `Waterbirds` for the best variant of proposed approach trained on `iNaturalist` and for methods such as 1-NN, ERM, and GroupDRO.

**Passing example groups as input.** In contrast to the findings in the single-task setting of Section 2, we observed that setting $\tilde{y}_i$ to represent group improves the proposed approach, even on top of permitting input dimensions and promoting induction heads. One case of this is presented in Figure 7, while more cases can be found in the complete results presented in Appendix B. For brevity, we mark passing groups as inputs with *+G* in figures and tables.

**Comparison with conventional learning algorithms.** Similar to the experiments in Section 2, we compare the best variant of the proposed approach (G + P + I) to 1-NN, ERM, and GroupDRO. Results presented in Figure 8 show that the learned algorithm is on-par with or outperforms the baselines starting at context length 32. The results at context lengths below 20 are not as informative, because the way we implemented the grafting operation implies that no examples are grafted when there are less than 10 examples in a class.

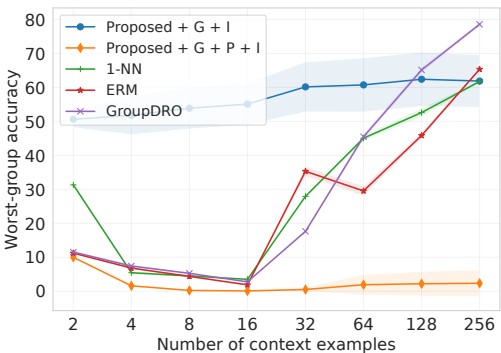

Figure 10: Worst-group test accuracy on `Waterbirds-severe` for the best variant of proposed approach trained on `iNaturalist` and for methods such as 1-NN, ERM, and GroupDRO.

**Generality of the learned algorithm.** To test the generality of the learned algorithm, we report evaluation results on `Waterbirds` (Figure 9) and `Waterbirds-severe` (Figure 10). We see that the learned algorithm outperforms baselines on `Waterbirds` and is as good as we got by training on `Waterbirds` itself. However, the learned algorithm fails completely on `Waterbirds-severe`, while the baselines give meaningful results starting at context length 32. We hypothesize that the challenge posed by the spurious features in `Waterbirds-severe` is significantly more severe compared to that in `iNaturalist`. Another likely cause is the mismatch between the ways that spurious features are encoded in embeddings. Interpolating between `Waterbirds` and `Waterbirds-severe` (by varying the norm of the added background vector), we find that there is a good generalization till the norm of the added vector is about 40% of the average embedding norm (see Figure 12 of Appendix B).

## 4. Discussion and conclusion

We have shown that it is possible to train an in-context learner tailored to one particular classification task with spurious features. To achieve this we introduced two key techniques: (a) permuting input embedding dimensions and (b) forming in-context learning sequences with intermediate queries simulating distribution shift. We have provided evidence that the learned algorithm is highly competitive on the task it was trained on. However, we found that while it generalizes to other tasks without spurious features, it does not work for tasks with other spurious features. Understanding this failure mechanistically and exploring techniques for enabling better generalization are key future research directions.

We next explored training on ICL synthetic instances of

diverse tasks and showed that it is possible to obtain an in-context learner that generalizes to unseen tasks, even with a different data generating process. We established the usefulness of two more techniques: (c) passing example groups as input and (d) promoting learning of induction heads by occasionally querying past context examples. We believe there is a room for improving in-context learning via improved strategies of choosing intermediate queries and possibly optimizing worst-group loss. Understanding why the learned algorithm fails under extreme distribution shifts and why variants with permutations fail more (see Figure 12) is an interesting question to explore. Another interesting direction to explore is to find out what exact algorithm is learned in the process of training on diverse tasks. Based on the results presented in this work, we conclude that the learned algorithm is neither 1-NN, ERM, or GroupDRO.

One ancillary finding of this work is that transformers can be trained to do in-context learning of classification tasks when good image embeddings are provided. This is remarkable because the input dimensionality we considered is much larger compared to that considered in the pioneering works of Garg et al. (2022) and Akyürek et al. (2022) (784 vs 20). Furthermore, we explored much larger context sizes (up to 512 examples instead of less than 100 examples) and observed improved performance with context size.

Our work has several important limitations. First of all, training a transformer-based in-context learner with high-dimensional image embeddings is computationally costly (see Appendix A for information on compute resources), although it is faster than the baselines at inference. For this reason, we did not explore more datasets and pretrained image embeddings. While we believe main conclusions of our work will be unchanged, the ranking of methods can vary with datasets and image embeddings. Indeed, we observed that 1-NN is unusually effective when applied on DINOv2 embeddings. Second, we compared in-context learners with limited number of methods among the multitude of them designed for robustness to spurious correlations. Third, we experimented with only one model size, width, and depth. Larger models might behave differently. Fourth, in our `iNaturalist` experiments, we considered only one "type" of spurious features. It is likely that this choice has significant effect on the learned algorithm and its generality. Future research should explore more ways of synthesizing spurious features and consider varying severity of the challenge posed by spurious features. The latter can be done by considering multiple spurious features, introducing label imbalance, varying magnitude of spurious correlations, and varying the margin spurious features provide. Finally, we acknowledge that the proposed approach requires spurious feature annotations which is typically costly to obtain. Fortunately, as we showed, it is possible to mitigate this limitation by creating synthetic data.

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

# A. Further experimental details

**Baselines.** For empirical risk minimization as a baseline, we tune 2 hyperparameters: learning rate (0.01 or 0.001) and number of epochs (100 or 200). For GroupDRO we additionally tune its parameter that controls adaptiveness of group weights (0.01, 0.1, or 1) and we also try an optional strong L2 regularization (1.0 weight decay), as it has been observed to be useful for small datasets (Sagawa* et al., 2020).

**Transformer-based methods.** In all transformer-based approaches, we train a causal decoder-only GPT-J transformer with 80M parameters that has 6 transformer layers with 8 multi-head attention, 768 model dimensionality, and 3072 hidden dimensionality. When training on `iNaturalist`, we add a layer normalization (Ba et al., 2016) on transformer input, as we expect input norms to change when we evaluate on `Waterbirds`-based datasets. The transformer input sequence in the proposed approach consists of 3 types of tokens: context image embeddings, query image embeddings, and label/group annotations. While the network can rely on positions and content to distinguish image embeddings from annotations, we found it to be helpful to encode token types explicitly. We do this by setting the first 3 dimensions of a token to be a one-hot vector representing token type (context image embedding, query image embedding, or annotation). When permuting dimensions, we do the permutation before encoding token types to keep the location of token-type information consistent. In our preliminary experiments and development, we used $n = 128$ context length. Apart from improved performance, we did not observe significant qualitative differences when we switched to larger context lengths for final experiments.

**Evaluation and model selection.** For all transformer-based approaches and baselines, we do a grid search to find the best combination of hyperparameters. In particular, we train each configuration with 5 different random seeds and selected one with the highest average test performance. Importantly, for baseline methods model selection is done for each context length independently, while for transformer-based methods model selection is done once with respect to the test performance at maximum context length observed during training. All evaluations are done on 8192 sequences, where the first $n$ examples are sampled from the corresponding train set while the query is sampled from the test set with a balanced group distribution. Finally, even when training transformers on permuted image embeddings, we do not apply permutations during evaluation. In all figures throughout this work, shaded regions show standard deviation across the 5 training runs.

Note that the most principled model selection approach would be selecting models based on a metric calculated on a dataset similar to the training set (e.g., a held-out part of training set), rather than the test set. For example, in the case of experiments on `Waterbirds` or `Waterbirds-severe`, the principled approach would be to select based on performance on sequences where the context part is sampled from the training set, while the final query is sampled from a held-out validation set with balanced group distribution. We tried this way of model selection and did not observe significant changes. In the case of experiments on `iNaturalist`, the principled approach would be to select based on performance on sequences where the context part is sampled from the training set, while the final query is sampled from the hold-out part the training set. We observed that this in-distribution metric is always around 99.5%-100%, and can be non-informative for model selection. This is a typical scenario in OOD generalization (see for example (Gulrajani & Lopez-Paz, 2021) or (Wenzel et al., 2022)).

**Definitions of metrics.** Given a set of predictions on `Waterbirds` or `Waterbirds-severe`, worst-group accuracy is defined as the lowest accuracy of predictions among the 4 groups. Note that worst-group accuracy is not applicable to `iNaturalist`, as different ICL sequences correspond to different classification tasks and hence form different groups. For this reason, we introduce minority-group and majority-group accuracies. Given a triplet $(C, q, \widehat{y})$, where $C$ is a context, $q$ is query, and $\widehat{y}$ is a prediction on $q$, we call $\widehat{y}$ a minority (majority) prediction, if $q$ is among the least (most) represented group(s) of the context $C$. Given a list of triplets $(C, q, \widehat{y})$, we define minority (majority) group accuracy as the accuracy among minority (majority) predictions.

**Compute resources.** We used NVIDIA A100 GPUs with 40GB memory to train transformer-based methods. The network we considered is small enough to fit on one GPU with batch size 32 when $n = 400$ (`iNaturalist` experiments) and batch size 24 when $n = 512$ (`Waterbirds` and `Waterbirds-severe` experiments). We did mixed 16-bit training to save compute and did not notice any quality degradation. A single training takes around 12 hours for `iNaturalist` experiments and around 18 hours for `Waterbirds` experiments. We used a mix of CPUs and weaker GPUs to train baselines, as they are not computationally as demanding.

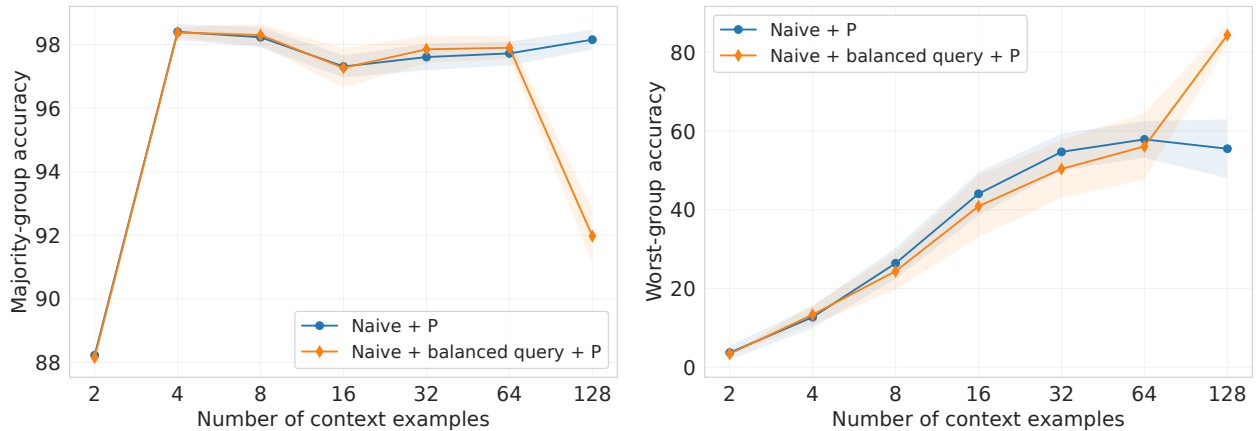

Figure 11: Majority-group and worst-group test accuracies on `Waterbirds-severe` as a function of context size for the naive approach with a single modification of making the last example (query) group-balanced. As expected, at intermediate context lengths this method performs similar to the naive approach, but is much better at the training context length.

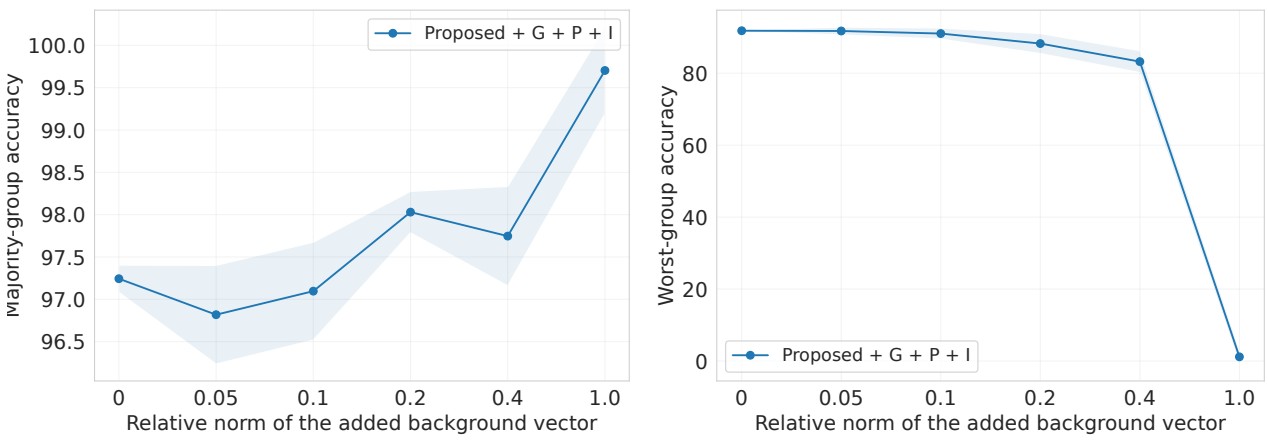

Figure 12: Majority-group and worst-group test accuracies of a proposed model (G + P + I) trained on `iNaturalist`, but evaluated on a modified variants of `Waterbirds` where we add a vector representing the spurious feature (background). The x-axis is the relative norm of the added vector compared to the average `Waterbirds` image embedding norm. Relative norm of 0 corresponds to `Waterbirds`, while relative norm of 1 corresponds to `Waterbirds-severe`.

## B. Additional results

In addition to the figures presented in the main text, here we provide the exact experimental resources for multiple transformer-based and baseline approaches, some of which were not included in the main text due to space constraints. Recall that +P means permuting input dimensions, +I means promoting learning of induction heads, and +G means passing example groups as input to in-context learning transformers.

Table 1 presents worst-group accuracies on the test set of `Waterbirds` for 3 sets of approaches: (a) in-context learners trained on `Waterbirds` itself, (b) in-context learners trained on `iNaturalist`, and (c) baselines. Similarly, Table 2 presents worst-group accuracies on the test set of `Waterbirds-severe` for 3 sets of approaches: (a) in-context learners trained on `Waterbirds-severe` itself, (b) in-context learners trained on `iNaturalist`, and (c) baselines. As RoPE-based transformers are not good at length extrapolation (Press et al., 2021), we do not attempt evaluating models trained on `iNaturalist` with context size 400 on 512-long sequences of `Waterbirds` or `Waterbirds-severe`. Finally, Table 3 presents minority-group accuracy on out-of-distribution classes of `iNaturalist` for two sets of approaches: (a) in-context learners trained on `iNaturalist` itself and (b) baselines.

Table 1: Complete results on `Waterbirds`. Reported numbers are average worst-group test accuracies, along with the their standard deviation. The top half of in-context learners were trained on `Waterbirds` itself, while the ones in the bottom half were training on `iNaturalist`.

| METHOD / CONTEXT SIZE | 4 | 8 | 16 | 32 | 64 | 128 | 256 | 512 |
|---|---|---|---|---|---|---|---|---|
| NAIVE | 87.02 (0.79) | 84.52 (1.00) | 85.14 (0.42) | 84.82 (0.89) | 83.41 (0.75) | 84.45 (1.04) | 85.08 (1.15) | 84.82 (1.26) |
| NAIVE + P | 70.92 (1.18) | 75.32 (1.11) | 80.66 (0.68) | 83.24 (0.35) | 86.87 (0.62) | 89.87 (0.85) | 91.94 (0.75) | 92.60 (0.59) |
| PROPOSED | 87.91 (1.29) | 85.63 (2.20) | 86.51 (2.17) | 85.42 (1.73) | 85.12 (2.34) | 85.86 (2.22) | 86.72 (1.89) | 86.89 (1.82) |
| PROPOSED + I | 88.18 (1.07) | 85.89 (1.31) | 86.68 (1.02) | 86.01 (1.39) | 84.82 (1.02) | 85.92 (1.23) | 86.07 (1.27) | 86.46 (1.57) |
| PROPOSED + P | 68.44 (2.40) | 73.46 (2.53) | 80.00 (2.06) | 83.71 (2.15) | 87.30 (1.79) | 90.02 (1.16) | 92.11 (1.65) | 91.95 (1.20) |
| PROPOSED + P + I | 68.05 (1.51) | 72.47 (1.80) | 78.97 (1.12) | 82.58 (0.68) | 86.39 (0.69) | 90.00 (0.60) | 91.78 (0.56) | 92.17 (0.86) |
| PROPOSED + G | 88.74 (1.01) | 87.00 (1.60) | 87.62 (1.58) | 86.86 (1.31) | 86.18 (1.33) | 86.91 (0.98) | 87.26 (1.11) | 86.95 (1.21) |
| PROPOSED + G + I | 88.89 (0.53) | 87.49 (0.69) | 87.70 (0.74) | 86.90 (0.95) | 86.03 (0.71) | 86.64 (0.72) | 87.29 (0.77) | 87.35 (1.00) |
| PROPOSED + G + P | 68.47 (2.32) | 73.74 (2.00) | 79.21 (1.68) | 82.85 (1.33) | 86.55 (1.17) | 89.98 (0.72) | 92.00 (0.82) | 93.05 (0.40) |
| PROPOSED + G + P + I | 68.24 (1.88) | 73.78 (1.67) | 80.23 (0.94) | 83.02 (1.22) | 86.94 (1.31) | 89.89 (0.91) | 92.46 (1.00) | 92.69 (1.15) |
| 1-NN | 65.29 (1.23) | 72.53 (1.11) | 79.15 (1.16) | 82.81 (0.63) | 87.49 (1.18) | 90.00 (1.05) | 91.96 (0.51) | 93.40 (0.27) |
| ERM | 63.04 (1.22) | 70.76 (1.01) | 77.32 (1.16) | 83.04 (1.09) | 85.95 (1.38) | 87.20 (0.77) | 88.10 (0.98) | 88.48 (0.45) |
| GROUPDRO | 64.61 (1.79) | 71.52 (0.73) | 77.81 (1.19) | 83.45 (1.57) | 87.34 (1.42) | 88.30 (0.91) | 89.79 (0.81) | 91.12 (0.62) |
| NAIVE | 69.77 (1.37) | 77.98 (1.51) | 79.23 (0.83) | 81.20 (1.35) | 82.57 (1.52) | 83.85 (1.56) | 84.21 (1.19) | - |
| NAIVE + P | 66.47 (1.17) | 73.12 (1.44) | 77.85 (1.74) | 81.76 (1.49) | 86.36 (0.86) | 88.02 (1.25) | 89.68 (0.77) | - |
| PROPOSED | 69.75 (5.51) | 77.51 (3.01) | 79.20 (2.11) | 81.39 (1.49) | 82.04 (1.29) | 83.51 (0.97) | 84.63 (0.80) | - |
| PROPOSED + I | 70.73 (1.42) | 77.10 (1.76) | 78.90 (1.49) | 80.86 (1.74) | 82.22 (1.72) | 84.22 (1.45) | 84.69 (1.47) | - |
| PROPOSED + P | 66.09 (1.49) | 73.71 (1.17) | 78.33 (0.69) | 82.75 (0.83) | 86.32 (0.52) | 88.85 (0.72) | 89.98 (1.35) | - |
| PROPOSED + P + I | 65.51 (2.16) | 70.91 (2.32) | 75.94 (3.04) | 81.51 (1.90) | 86.41 (1.50) | 89.39 (0.98) | 91.08 (0.75) | - |
| PROPOSED + G | 70.98 (2.52) | 78.41 (1.25) | 79.67 (1.26) | 81.59 (1.42) | 82.42 (1.28) | 83.91 (1.64) | 84.31 (1.31) | - |
| PROPOSED + G + I | 71.94 (2.70) | 78.56 (1.65) | 80.62 (1.66) | 82.31 (1.76) | 83.52 (1.57) | 84.52 (1.32) | 85.35 (1.20) | - |
| PROPOSED + G + P | 67.55 (0.78) | 73.79 (0.33) | 78.32 (0.93) | 82.56 (1.31) | 86.01 (1.09) | 89.40 (1.22) | 90.99 (1.15) | - |
| PROPOSED + G + P + I | 69.18 (2.76) | 74.13 (2.06) | 79.18 (1.81) | 83.17 (0.85) | 87.25 (0.37) | 90.67 (0.80) | 92.23 (0.69) | - |

Table 2: Complete results on `Waterbirds-severe`. Reported numbers are average worst-group test accuracies, along with the their standard deviation. The top half of in-context learners were trained on `Waterbirds-severe` itself, while the ones in the bottom half were training on `iNaturalist`.

| METHOD / CONTEXT SIZE | 4 | 8 | 16 | 32 | 64 | 128 | 256 | 512 |
|---|---|---|---|---|---|---|---|---|
| NAIVE | 83.04 (1.92) | 80.78 (1.58) | 80.78 (1.85) | 79.43 (2.77) | 80.50 (2.43) | 80.29 (2.30) | 81.67 (2.25) | 82.02 (2.72) |
| NAIVE + P | 10.89 (2.71) | 28.61 (4.98) | 46.23 (4.17) | 58.40 (2.46) | 67.13 (2.34) | 74.28 (2.25) | 77.18 (3.11) | 77.49 (4.08) |
| PROPOSED | 82.64 (1.56) | 81.01 (2.23) | 81.90 (1.80) | 81.36 (1.69) | 81.94 (1.91) | 81.70 (1.62) | 82.35 (1.72) | 82.09 (2.15) |
| PROPOSED + I | 83.23 (1.30) | 80.76 (1.93) | 81.65 (2.38) | 81.46 (2.11) | 81.63 (2.40) | 81.34 (2.01) | 81.46 (2.32) | 82.24 (3.49) |
| PROPOSED + P | 61.94 (8.91) | 68.23 (5.53) | 75.94 (3.13) | 81.93 (1.53) | 85.76 (2.03) | 88.36 (1.30) | 90.01 (1.98) | 90.20 (2.65) |
| PROPOSED + P + I | 64.01 (4.05) | 72.22 (4.43) | 78.45 (2.79) | 82.00 (2.20) | 85.86 (1.64) | 88.13 (1.39) | 90.09 (1.73) | 90.59 (1.54) |
| PROPOSED + G | 82.02 (3.37) | 81.15 (3.56) | 83.11 (1.84) | 81.22 (2.08) | 81.30 (1.72) | 81.90 (1.62) | 82.48 (1.62) | 82.44 (1.34) |
| PROPOSED + G + I | 82.61 (3.42) | 80.48 (2.69) | 81.20 (3.55) | 80.13 (3.18) | 81.09 (2.86) | 80.84 (2.47) | 81.61 (2.36) | 81.84 (2.51) |
| PROPOSED + G + P | 59.11 (2.89) | 64.44 (5.67) | 71.30 (3.74) | 79.46 (0.83) | 85.21 (1.54) | 88.60 (1.36) | 90.65 (1.01) | 91.38 (1.14) |
| PROPOSED + G + P + I | 64.26 (5.81) | 70.05 (4.01) | 77.76 (1.77) | 82.38 (1.66) | 86.56 (0.88) | 89.09 (1.02) | 90.75 (0.96) | 90.82 (0.73) |
| 1-NN | 5.44 (0.60) | 4.50 (0.43) | 3.49 (0.21) | 27.92 (0.54) | 45.04 (0.88) | 52.58 (1.39) | 61.74 (0.48) | 71.20 (0.58) |
| ERM | 6.81 (0.44) | 4.35 (0.26) | 1.87 (0.24) | 35.30 (1.55) | 29.52 (1.49) | 45.84 (1.00) | 65.35 (0.53) | 75.69 (0.88) |
| GROUPDRO | 7.42 (0.57) | 5.26 (0.35) | 2.75 (0.29) | 17.62 (0.65) | 45.47 (1.18) | 65.13 (1.06) | 78.57 (0.77) | 86.89 (0.57) |
| NAIVE | 48.18 (3.52) | 49.39 (3.28) | 48.71 (6.49) | 52.58 (4.56) | 54.10 (6.04) | 56.41 (5.03) | 56.86 (4.75) | - |
| NAIVE + P | 0.88 (0.45) | 0.06 (0.05) | 0.00 (0.00) | 0.13 (0.29) | 0.19 (0.43) | 0.13 (0.29) | 0.02 (0.04) | - |
| PROPOSED | 49.04 (2.76) | 53.39 (4.74) | 54.82 (8.82) | 59.44 (10.75) | 61.04 (12.23) | 62.26 (12.31) | 63.77 (12.38) | - |
| PROPOSED + I | 48.45 (6.15) | 52.44 (11.15) | 54.74 (10.69) | 58.67 (11.38) | 60.37 (9.19) | 62.42 (8.69) | 63.27 (9.15) | - |
| PROPOSED + P | 1.88 (0.56) | 0.27 (0.21) | 0.06 (0.10) | 0.08 (0.13) | 0.30 (0.60) | 0.15 (0.28) | 0.03 (0.06) | - |
| PROPOSED + P + I | 2.27 (0.74) | 0.66 (0.49) | 0.15 (0.14) | 1.18 (1.09) | 2.50 (2.49) | 1.14 (0.88) | 0.50 (0.20) | - |
| PROPOSED + G | 50.00 (5.03) | 52.31 (5.05) | 53.69 (4.54) | 57.87 (3.33) | 59.11 (3.16) | 60.33 (3.36) | 62.30 (3.01) | - |
| PROPOSED + G + I | 51.78 (5.76) | 53.87 (6.15) | 55.07 (6.20) | 60.15 (7.40) | 60.73 (8.02) | 62.40 (8.01) | 61.86 (7.77) | - |
| PROPOSED + G + P | 1.52 (0.69) | 0.16 (0.13) | 0.00 (0.00) | 0.10 (0.20) | 0.04 (0.05) | 0.03 (0.05) | 0.36 (0.73) | - |
| PROPOSED + G + P + I | 1.59 (0.17) | 0.23 (0.16) | 0.08 (0.10) | 0.50 (0.69) | 1.91 (3.05) | 2.19 (3.67) | 2.34 (4.00) | - |

Table 3: Complete results on `iNaturalist`. Reported numbers are average minority-group accuracies on the OOD test set of `iNaturalist`, along with the their standard deviation.

| Method / Context size | 4 | 8 | 16 | 32 | 64 | 128 | 256 | 400 |
|---|---|---|---|---|---|---|---|---|
| PROPOSED | 91.80 | 93.20 | 93.71 | 94.58 | 95.01 | 95.27 | 95.30 | 94.89 |
| | (0.39) | (0.29) | (0.35) | (0.22) | (0.42) | (0.42) | (0.40) | (0.27) |
| PROPOSED + I | 92.88 | 93.82 | 94.61 | 95.36 | 95.76 | 95.90 | 95.94 | 95.06 |
| | (0.31) | (0.37) | (0.56) | (0.45) | (0.40) | (0.44) | (0.18) | (0.54) |
| PROPOSED + P | 92.04 | 92.90 | 94.80 | 96.64 | 97.65 | 98.39 | 98.49 | 98.55 |
| | (0.22) | (0.30) | (0.32) | (0.30) | (0.20) | (0.27) | (0.14) | (0.23) |
| PROPOSED + P + I | 92.15 | 92.97 | 94.67 | 96.86 | 97.80 | 98.46 | 98.54 | 98.61 |
| | (0.28) | (0.30) | (0.28) | (0.21) | (0.29) | (0.20) | (0.11) | (0.25) |
| PROPOSED + G | 92.48 | 93.27 | 93.88 | 94.91 | 94.99 | 95.29 | 95.13 | 94.64 |
| | (0.45) | (0.72) | (0.43) | (0.63) | (0.38) | (0.45) | (0.33) | (0.43) |
| PROPOSED + G + I | 92.59 | 93.80 | 94.18 | 95.50 | 95.82 | 95.83 | 95.82 | 95.28 |
| | (0.33) | (0.23) | (0.38) | (0.33) | (0.41) | (0.34) | (0.55) | (0.60) |
| PROPOSED + G + P | 91.90 | 92.84 | 94.69 | 97.28 | 98.29 | 98.70 | 98.85 | 99.00 |
| | (0.17) | (0.19) | (0.15) | (0.31) | (0.13) | (0.19) | (0.19) | (0.11) |
| PROPOSED + G + P + I | 92.28 | 93.25 | 94.93 | 97.73 | 98.44 | 98.99 | 99.04 | 99.06 |
| | (0.10) | (0.09) | (0.22) | (0.07) | (0.20) | (0.09) | (0.14) | (0.07) |
| 1-NN | 92.08 | 94.56 | 95.84 | 97.17 | 97.84 | 98.49 | 98.55 | 98.80 |
| | (0.64) | (0.39) | (0.16) | (0.23) | (0.12) | (0.20) | (0.23) | (0.21) |
| ERM | 89.67 | 92.98 | 94.65 | 96.17 | 96.88 | 97.70 | 98.15 | 98.43 |
| | (0.43) | (0.30) | (0.17) | (0.24) | (0.23) | (0.21) | (0.17) | (0.11) |
| GROUPDRO | 91.20 | 93.79 | 95.33 | 97.39 | 97.85 | 98.46 | 98.91 | 99.01 |
| | (0.55) | (0.39) | (0.18) | (0.20) | (0.20) | (0.13) | (0.20) | (0.18) |

