# OpenReview forum: "In-context learning in presence of spurious correlations"
_ICML.cc/2024/Workshop/ICL — ICML 2024 Workshop ICL Poster_

### Official Review · Reviewer_VfpV · 2024-06-08
**Interesting perspective of ICL on image data with spurious correlations**

**Rating:** 2
**Fit:** 3
**Confidence:** 2

**Workshop Review:**

Summary: this paper focuses on ICL on image data in the presence of spurious features. It first focuses on a single task, showing that naive ICL is prone to learning the spurious features and that permuting the images and including intermediate queries can help mitigate this problem. It then moves on to a multi-task setting, showing that passing example groups as input and replacing some queries with preceding ones can further promote ICL.

Clarity: paper is quite hard to follow at times and can be further improved by providing more concrete examples and motivations (especially in Section 3).

Correctness: results generally seem reasonable, although it is good to include some motivation/intuition as to why the proposed techniques improve ICL.

Novelty: this work is novel enough, although its applicability can be further discussed (e.g., in reality, should we use the proposed techniques when constructing prompts for visual ICL?)

Interest to the community: this work is of wide interest to the ICL and general ML community.

**Reason For Not Giving Higher Score:**

Writing is not clear at times. Also, this paper generally lacks motivation and explanations.

**Reason For Not Giving Lower Score:**

This paper provides an interesting and novel viewpoint of ICL under spurious correlations. I recommend acceptance (hence a rating of 2 and not 1), provided that the authors can improve the organization of the paper and provide sufficient motivation/justifications of the proposed approaches.

---

### Official Review · Reviewer_vKKu · 2024-06-09
**The paper analyzes in-context learning with respect to datasets containing spurious correlations. While the paper presents an interesting setup, the method can be largely improved to make it more generic and adaptable.**

**Rating:** 1
**Fit:** 3
**Confidence:** 2

**Workshop Review:**

Strengths.
1. The paper is well written.
2. The setting of in-context learning for tasks containing spurious correlations is novel and interesting.

Weaknesses.
1. I feel that the proposed approach is very data-driven. Thus, I am not convinced of its utility across benchmarks and tasks. There are multiple benchmarks, such as CelebA, subpopulation shift benchmark, WILDS, and DomainBed.
2. A rigorous related works section would be helpful. I know of multiple relevant papers in the same setting, which the authors have not mentioned or even compared with. As an example, a) Context is Environment (Gupta et al. 2023) b) Contextual Vision Transformers for Robust Representation Learning (Bao et al. 2023)

**Reason For Not Giving Higher Score:**

While the paper presents an interesting problem setting, the approach is very data-driven and not generalizable. For instance, merely applying permutations to image embeddings is not well-motivated and justified. Further, intuitions from the work are missing. For instance, if the context is informative enough to provide some "z" that helps improve predictions from x to y, training would benefit from context, and the model will use the context to improve predictions. Instead of simply permutating image embeddings to ensure the model uses context, more principled ways will be better and generalizable.

**Reason For Not Giving Lower Score:**

N/A

---

### Meta-Review · Area_Chair_n7wx · 2024-06-17

**Recommendation:** 2

**Metareview:**

A reviewer had concerns with the ad hoc nature of the task and the evaluation is limited, but the investigations in this paper into the behavior of ICL methods with spurious features are interesting nonetheless.

---

### Decision · Program_Chairs · 2024-06-17

Accept (Poster)